# Proposal for Some Affordable Laboratory Biofilm Reactors and Their Critical Evaluations from Practical Viewpoints

**DOI:** 10.3390/ma15134691

**Published:** 2022-07-04

**Authors:** Hikonaru Kudara, Hideyuki Kanematsu, Dana M. Barry, Akiko Ogawa, Takeshi Kogo, Hidekazu Miura, Risa Kawai, Nobumitsu Hirai, Takehito Kato, Michiko Yoshitake

**Affiliations:** 1National Institute of Technology (KOSEN), Suzuka College, Suzuka 510-0294, Japan; s18m172@kagawa-u.ac.jp (H.K.); ogawa@chem.suzuka-ct.ac.jp (A.O.); kougo@mse.suzuka-ct.ac.jp (T.K.); kawai-r@mse.suzuka-ct.ac.jp (R.K.); hirai@chem.suzuka-ct.ac.jp (N.H.); 2Department of Electrical & Computer Engineering, Clarkson University, Potsdam, NY 13699, USA; dmbarry@clarkson.edu; 3STEM Laboratory, State University of New York, Canton, NY 13617, USA; 4Faculty of Medical Engineering, Suzuka University of Medical Science, Suzuka 510-0293, Japan; miura-h@suzuka-u.ac.jp; 5National Institute of Technology (KOSEN), Oyama College, Oyama 323-0806, Japan; kato_t@oyama-ct.ac.jp; 6National Institute for Materials Science (NIMS), Tsukuba 305-0044, Japan; yoshitake.michiko@nims.go.jp

**Keywords:** biofilms, laboratory biofilm reactors, LBR

## Abstract

Biofilms are a result of bacterial activities and are found everywhere. They often form on metal surfaces and on the surfaces of polymeric compounds. Biofilms are sticky and mostly consist of water. They have a strong resistance to antimicrobial agents and can cause serious problems for modern medicine and industry. Biofilms are composed of extracellular polymeric substances (EPS) such as polysaccharides produced from bacterial cells and are dominated by water at the initial stage. In a series of experiments, using Escherichia coli, we developed three types of laboratory biofilm reactors (LBR) to simulate biofilm formation. For the first trial, we used a rotary type of biofilm reactor for stirring. For the next trial, we tried another rotary type of reactor where the circular plate holding specimens was rotated. Finally, a circular laboratory biofilm reactor was used. Biofilms were evaluated by using a crystal violet staining method and by using Raman spectroscopy. Additionally, they were compared to each other from the practical (industrial) viewpoints. The third type was the best to form biofilms in a short period. However, the first and second were better from the viewpoint of “ease of use”. All of these have their own advantages and disadvantages, respectively. Therefore, they should be properly selected and used for specific and appropriate purposes in the future.

## 1. Introduction

Problems caused by biofilms are found in various fields. Since biofilms form on materials’ surfaces, some interactions between materials and bacterial environments must be related to the formation and growth of biofilms. The concept of biofilms was proposed from the late 1970s to 1980s [1,2] by medical [3] and environmental scientists [4,5].Biofilms usually form on solid materials. Therefore, it is very important to develop antibiofilm materials [6,7]. To achieve this goal, we need to determine the mechanism for biofilm formation and growth and the factors involved. This requires appropriate evaluation processes (optical microscopes and electron microscopes [8,9], scanning electron microscopes [10,11,12], confocal laser microscopy [13,14,15,16,17], FTIR [18,19,20,21], and Raman spectroscopy [22,23,24,25]). When we began our investigations about biofilms from the viewpoint of materials science [26,27,28,29,30,31,32,33], we did not have any appropriate and well-known biofilm evaluation processes to use. Although many methods related to biology, environmental science, and medical science were available, we needed some appropriate processes to evaluate biofilms on materials. Our evaluation process is composed of two steps. The first step is biofilm formation. Biofilm reactors are used to make biofilms, which might be natural or artificial. Particularly, biofilm reactors used at the laboratory scale are called a laboratory biofilm reactors (LBRs). Then quantitative or qualitative evaluation of biofilms on materials produced in laboratory biofilm reactors (LBRs) is needed. The second step involves The Society of International Sustaining Growth for Antimicrobial Articles (SIAA) in Japan and their current plans to establish an International Standard to evaluate biofilms on materials. However, globally, reactors of the type in the first step have not been considered for standardization yet. Therefore, we must continue our investigations and trials, so that practical engineers can use biofilm reactors as a common evaluation tool. So far, we have developed some laboratory biofilm reactors.

This paper mentions biological researchers and engineers who have used biofilm reactors where the flow factors have not been considered. However, the flow should be incorporated into the artificial production phase of biofilms since biofilms often form on materials’ surfaces in fluid environments. For example, consider scale formation in bathtubs, kitchen sinks, and various water and sewage pipes. To simulate these environments, the conditions will differ from case to case. Using common reactors for these items is impossible. On the contrary, we need to devise a biofilm reactor that is simple, intuitive, and practical, as well as applicable to as many cases as possible. We devised three types of biofilm reactors for use at the laboratory scale, where flow factors can be incorporated to some extent and affordable to practical researchers and engineers at the same time. In this paper, we compared them from the practical viewpoint and mentioned problems, citing some concrete examples.

## 2. Experimental

### 2.1. Proposals for the New Artificial Laboratory Biofilm Reactors and Their Concepts

We arranged and made a rotary LBR using a magnetic stirrer, so that the culture part and biofilm formation (the specimen part) within the LBR can share a common space. In the past, we developed a flow-type biofilm reactor, where the incubation of bacteria was placed apart from the biofilm formation part. In this case, biofilm formation was accelerated because biofilm growth and bacterial growth were close to each other. A schematic diagram of this is shown in Figure 1. In this reactor, the solution was circulated within the reactor and circulation was caused by the stirrer. Therefore, we tentatively call this the stirrer-driven rotary biofilm reactor (SDRBR). The SDRBR is composed of three-neck flasks (500 mL), a magnetic stirrer, and a fixation jig. The jig is made of metallic materials (stainless steel) and inserted into the flask. The silicon rubber was used at the inlet part to fix the jig where specimens were attached. The jig with a specimen and a stirrer were inserted into the three-neck flask. In the SDRBR, only one specimen can be used in each experiment.

On the other hand, we designed and made the other rotation type of LBR. In the rotary LBR with a rotating jig, we developed an LBR in which the area where *E. coli* is cultured and the area where the sample exists are the same as in the rotary LBR with a stirrer. In the rotating jig, a jig to which various samples are fixed, it rotates by coupling with a motor. By this mechanism, the jig has the function of fixing the samples and agitating them. Figure 2 shows the setup. We named this type of LBR: a rotating-platform-driven LBR (RPDLBR).

The third type is tentatively called the closed-loop circulation LBR (CLC LBR) by the authors [34]. Polymeric tubes (Suffeed tubes, TERUMO Co., Tokyo, Japan) were connected to a 500 mL^3^-neck beaker (SCHOTT DURAN, Jena, Germany), and an acrylic column containing a sample fixed with an acrylic jig (in the center) was incorporated through silicone rubber. A peristaltic pump (tubing pump) was incorporated into the Suffeed connecting tube to create a circulating LBR. The circulating LBR was placed in a table-top clean booth (AS ONE) or in an incubator during the experiment.

The laboratory biofilm reactors, including the CLC LBR used in this study, and their experimental conditions are summarized schematically in Figure 3.

### 2.2. Biofilm Formation and Evaluations Used as Comparative Examples

#### 2.2.1. Specimens and Bacteria

In this experiment, we concentrated on the differences between the biofilm reactors. We considered their effects on the evaluation results and the various characteristics of each setup (apparatus). Commercial pure metals of titanium and aluminum were used in this experiment. Specimens of these metals do not easily form biofilms and avoid the formation of corrosion products (in our experimental conditions) due to the inherit dense oxide films.

For a source of bacteria, we used *E. coli* (K12 G6). We have often used this kind of bacteria for experiments and can fix biofilm formation. Therefore, data in this experiment can be compared with those from previous experiments.

#### 2.2.2. Preparation of Bacterial Solution and the Biofilm Formation Process

An amount of 12.5 g of LB medium was added to 500 mL of distilled water, stirred for 5 min to dissolve, and autoclaved at 121 °C/15 min. A volume of 500 mL of liquid LB medium was used as a solution for the rotary LBR and circulating LBR stirrer. An amount of 25 g of LB medium was added to 1000 mL of distilled water, stirred for 5 min to dissolve, and autoclaved at 121 °C/15 min. A volume of 1000 mL of liquid LB medium, which was dissolved by stirring and sterilized in an autoclave at 121 °C/15 min, was used as the solution for the rotary LBR with a stirrer. The K12 strain of *E. coli* was used. The bacteria were also cultured successively on LB agar medium. As a pre-culture before the experiment, one colony of *E. coli* (K12) was taken from LB agar medium in a loop and placed in a test tube containing 200 mL of undiluted LB medium and incubated for 18 h. Additionally, then, they were used as bacterial solution.

Various samples were inserted into the jig, which was then bonded to various LBRs. To make the inside of each LBR sterile, the LBR was sealed and autoclaved at 121 °C for 20 min. After undergoing pressure sterilization, 500 μm of pre-cultured *E. coli* was placed in the rotating LBR and circulating LBR using a stirrer and 1000 μm in the rotating LBR (using a rotating jig in a clean bench). This resulted in a concentration of 1 mL/1000 mL in the liquid LB medium of the culture medium in which *E. coli* was cultured. Biofilm formation was performed by operating the various LBRs. The temperature during the experiment was set at 25 °C and the duration of the experiment was 24 h.

### 2.3. Evaluation of Biofilms

To evaluate biofilms, we used two kinds of evaluation methods. One of them was Raman spectroscopy and the other was crystal violet (CV) staining.

#### 2.3.1. Raman Spectroscopy

Biofilms formed on specimens were freeze-dried to fix the components of biofilms before Raman spectroscopy. Ninety percent of the biofilm’s constituents is H_2_O. Therefore, if the sample is left in the air, the H_2_O evaporates and the biofilm shrinks, changing its structure. To prevent this, freeze-drying was used to solidify the biofilm formed on the sample’s surface as a post-experiment sample treatment. This method made it possible for us to avoid cases where planktonic bacteria and polymeric substances derived only from LB media (irrelevant of biofilms) are detected during the evaluation process.

For the freeze-drying procedure, water, ethanol, and t-butyl alcohol were prepared as solutions. First, solutions were prepared by mixing water and ethanol in the following ratios: 7:3, 5:5, 3:7, 2:8, 1:9, 0.5:9.5, 0.2:9.8, and 0:1. The samples were immersed in the solutions of each concentration for 15 min, from left to right, as the concentration of ethanol increased. Next, ethanol: t-butyl alcohol solutions were prepared in the proportions 7:3, 5:5, 3:7, and 0:1, and the samples were immersed in each solution for 15 min, starting from the left, as the concentration of t-butyl alcohol increased. The samples were then placed in a freezer for at least 30 min to freeze. A vacuum pump was used to evacuate the frozen samples.

A laser Raman spectrophotometer (NRS-3100, JASCO Co., Tokyo, Japan) was used. Raman spectroscopy is a method of analyzing molecular structure by irradiating a sample with laser light and analyzing Raman scattering, which is extremely weak compared to Rayleigh scattering. This method is used to analyze and compare samples before and after experiments to analyze the various organic substances in the EPS in biofilms, mainly polysaccharides, nucleic acids, proteins, and lipids. Data analyzed using the Raman spectrophotometer were baseline corrected and smoothed. Peaks were identified, comparing the data obtained in our previous experiments and data from other researchers’ studies. Since the apparatus has its own optical microscope, we observed the materials’ surfaces by using the function and fixed the place of green laser irradiation (100 mW, 532 nm).

#### 2.3.2. Crystal Staining

Staining with crystal violet stains proteins and polysaccharides contained in the biofilm. First, 0.1% crystal violet is prepared, and the biofilm-formed specimen is immersed in it for 30 min, after which the specimen’s surface is rinsed with tap water. The color change in the material’s surface is evaluated by using a color meter (CR-13, Konika-Minolta Sensing Co. Ltd., Tokyo, Japan) and their L*, a* and b* values were used for the evaluation of colored surfaces by crystal violet solution. In this paper, we used L, a and b were used as conventional short technical term for L*, a* and b*. Figure 4 shows L-a-b color space and the positions/mutual relations schematically.

## 3. Results

### 3.1. Results from the Stirrer-Driven Rotary Biofilm Reactor (SDRBR)

The results of the Raman spectroscopic experiments performed in a rotating LBR with a stirrer are shown in Figure 5 and Figure 6. Figure 5 shows the specimens’ surfaces observed by the microscope and Figure 6 shows Raman peaks for titanium and aluminum specimens.

The shaded areas one observes by using optical microscopes often correspond to biofilms. Therefore, such areas were used as landmarks for observation as shown in Figure 5. The center of each optical microscopic image provides Raman peaks shown in Figure 6, respectively. Since peaks for titanium specimens were small, we presume biofilms in this case were not so remarkable. If we counted four tiny peaks as biofilm components in Figure 6a, the following four peaks can be mentioned: 1558 cm^−1^ (proteins such as amide II) [35], 1149 cm^−1^ (lipids) [36], 1053 cm^−1^ (lipids) [36] and 869 cm^−1^ (polysaccharides) [37]. We presume that these peaks show the existence of biofilms on titanium specimens. However, they can be derived from LB media. Even in such a case, the existence of organic matter shows that the surfaces are sticky and the stickiness obviously shows the existence of biofilms (since stickiness is generally caused by biofilms). From the practical viewpoint, the peaks of organic matter (even after washing and the substitution processes) show the existences of biofilms directly or indirectly. Figure 6b shows Raman peaks observed on the aluminum specimen’s surface where the location corresponds to Figure 5b. Figure 6b shows a better view of the Raman peaks at the following locations: 1560 cm^−1^ (amide II) [35], 1490 cm^−1^ (protein) [35], 1430 cm^−1^ (lipids) [36], 1129 cm^−1^ (lipids) [36,38] and 1197 cm^−1^ (lipids) [36,38]. These peaks were derived from biofilms directly or obtained from sticky surfaces caused by biofilms in the same way.

Figure 7 shows the results of staining by using 0.1% crystal violet solutions. L*a*b* values were measured by using the apparatus. To show the extent of staining into violet colors, the values of a* and b* were plotted in the a–b plane. The figures show that the plots of titanium and aluminum specimens moved from the original points to the staining ones. The point corresponding to a violet color is in the fourth quadrant. Both plots of titanium and aluminum tended to move from the first or the second quadrants to the fourth one with staining. The changes show the surfaces were stained by crystal violet due to the existence of biofilms. Additionally, the length of change in the space corresponds to the extent of staining. In this case, the extent of staining for the titanium specimen was smaller than that of the aluminum specimen.

### 3.2. Results from the Rotating-Platform-Driven LBR (RPDLBR)

The results of the experiments performed on a rotary LBR with a rotating fixture (RPDLBR) are shown in Figure 8 and Figure 9. Figure 8 shows the specimens’ surfaces observed by using the optical microscope. Figure 9 shows Raman peaks obtained for titanium and aluminum specimens.

The results of Raman spectroscopic analysis detected peaks of organic matter that may be of biological origin in the various samples. Therefore, the surface adherends observed by using optical microscopy are considered to mainly be organic materials of biological origin. The following results indicate certain materials for the titanium specimens: 1326 cm^−1^ (Protein) [38], 1283 cm^−1^ (amide III or protein) [35,36], 1151 cm^−1^ (Lipids) [36], 977 cm^−1^ (nucleic acids or lipid) [35], 853 cm^−1^ (polysaccharides) [37] and 745 cm^−1^ (polysaccharides or lipids) [35]. Most of these materials might be derived from biofilms, even though some of them can come from floating organic compounds in the system. At any rate, the attached organisms show the existence of biofilms, as we already described. Keep in mind that the results displayed in Figure 6 show weak and not clearly defined peaks for the titanium specimens. This information suggests that the extent of biofilm formation is not be very large. On the other hand, the Raman peaks for the aluminum specimens were detected at 1326 cm^−1^ (lipid or protein) [35,38], 1283 cm^−1^ (amide III or lipid) [36,39], 1151 cm^−1^ (lipids) [36], 977 cm^−1^ (Nucleic acids or lipid) [35], 853 cm^−1^ (polysaccharides) [37] and 745 cm^−1^ (protein) [39]. These results suggest that biofilms were formed when the samples were observed locally. The crystal violet staining confirmed that all samples were stained purple, although there were differences in the staining.

Figure 10 shows the color plots of stained surfaces and their changes after biofilm formation. Like the results in Figure 7, the change in surface color shows the formation of biofilms. Even though both specimens showed a color change to blue, the extent was low for the titanium specimen. This was a common pattern in the results of Raman spectroscopy and color measurement. At the same time, this suggests that it is difficult to form biofilms on the specimen and this was similar between the SDRBR and the RPDLBR.

### 3.3. Results from Using the Closed-Loop Circulation LBR (CLC LBR)

The results of experiments with Ti and Al using a circulating LBR (closed-loop circulation LBR) are shown in Figure 11, Figure 12 and Figure 13. Figure 11 shows the dark areas which are supposed to biofilms. When the areas were irradiated by a laser beam, peaks were obtained, as shown in Figure 12. The results of Raman spectroscopic analysis (Figure 12) showed that peaks for the titanium specimen were at 1437 cm^−1^ (Lipids) [36,38], 1296 cm^−1^ (amide II or lipids) [35,36], 1125 cm^−1^ (Lipids) [38] and 1058 cm^−1^ (Lipids) [36]. These are considered to indicate biofilm origin. On the other hand, the Raman peaks for the aluminum specimen were at 1430 cm^−1^ (Lipids) [38], 1338 cm^−1^ (protein) [39], 1296 cm^−1^ (lipid or amid III) [36,38] and 1120–1185 cm^−1^ (Lipids and/or proteins) [38]. They also showed the existence of biofilms on the specimen. Compared with the results for the other two types of LBRs, Raman peaks were clearly seen, and their S/N ratios were relatively high, particularly for the titanium specimen. The reason can be attributed to the flow type for this apparatus. Figure 13 shows the color changes before and after biofilm formation. Additionally, in this case, the color change to violet can be confirmed. As for the titanium specimen, the color change to violet can be seen more clearly, as compared to the results for the RPDLBR.

## 4. Discussion

For all cases described above, biofilms can form on specimens to a greater or lesser extent. Even though generalization of the results might be difficult to determine in this experiment, we can evaluate the characteristics of three kinds of LBRs from the practical viewpoint. The difference also depends on the type of LBR used because each one has its own merits and limitations. Therefore, a certain LBR type should be selected for a specific purpose and application. We compared the characteristics of these LBRs. This information is summarized in Table 1.

From the practical viewpoint of social implementation, we analyze and compare the three types of laboratory biofilm reactors, as shown in Table 1. Biofilms are formed by using the three types of reactors. The extent of biofilm formation was a bit weak in the SDRBR and the RPDLBR as compared with the CLCBR. This suggests that parallel flow is effective at forming biofilms. The types of biofilm components depend on the type of LBR used. Proteins are the main component for the SDRBR and the RPDLBR, while lipids occupy biofilms in the CLCLBR. These results may be attributed to the ability of the biofilm components to remain on materials’ surfaces against the flow. Liquid flow can remove bacteria and some components of biofilms. Under these experimental conditions (the balance between the adherence force of components and flow strength), the results are shown in Table 1. The structure might limit the capacity of how many specimens can be treated at the same time. The SDRBR can deal with one or two specimens simultaneously. On the other hand, the RPDLBR and the CLCLBR can deal with a couple of specimens. However, more revisions for both types will improve their capacities. As for “ease of use”, the SDRBR was the best, followed by the RPDLBR and ending with the CLCLBR. The most difficult problem for this project has been sterilization of the devices. The larger the device, the more difficult sterilization becomes in many ways. This factor should be incorporated into further studies.

## 5. Conclusions

With pure titanium and aluminum specimens as model metallic materials, we carried out biofilm formation tests, using three different laboratory biofilm reactors that we designed and produced for practical purposes. These LBRs can produce the flow in the systems and can be applied to the practical acceleration tests for industries. They were named the stirrer-driven rotary biofilm reactor (SDRBR), the rotating-platform-driven LBR (RPDLBR) and the closed-loop circulation LBR (CLC LBR). The SDRBR and the RPDLBR belong to the same category because the rotating flow is the driving force to form biofilms. On the other hand, linear parallel flow is added to the specimens’ surfaces.

Closed-loop circulation LBRs are a little more difficult to handle, but this type can form biofilms the most effectively.

The rotating flow LBRs can be easily pressurized and sterilized, so they should be easy to handle for biofilm formation using bacteria. They are also considered to be easy to use in real-life, non-living environments because they are not large devices and can provide flow velocity to the sample.

Biofilm formation was observed by using our devised LBRs. However, each one should be selected for different/specific purposes, according to flow types and conditions for practical situations.

To develop anti-biofilm materials and for their societal implementation in the future, these models should be improved further. However, prototypes such as shown in these experiments will be useful and good references, when each is properly selected and used for specific and appropriate purposes.

## Figures and Tables

**Figure 1 materials-15-04691-f001:**
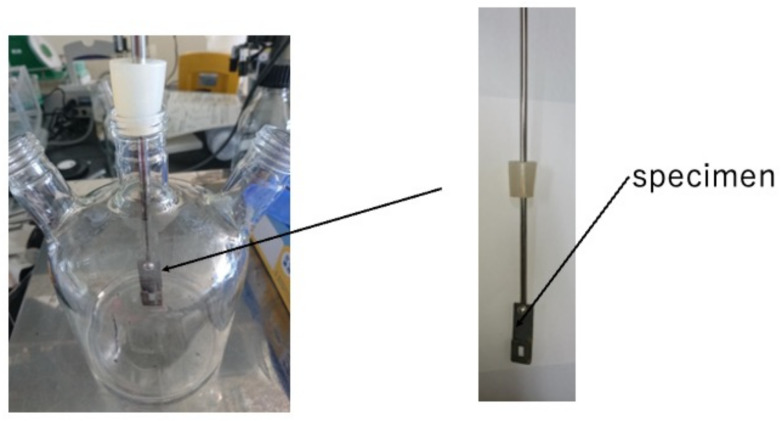
Stirrer-driven rotary biofilm reactor (SDRBR).

**Figure 2 materials-15-04691-f002:**
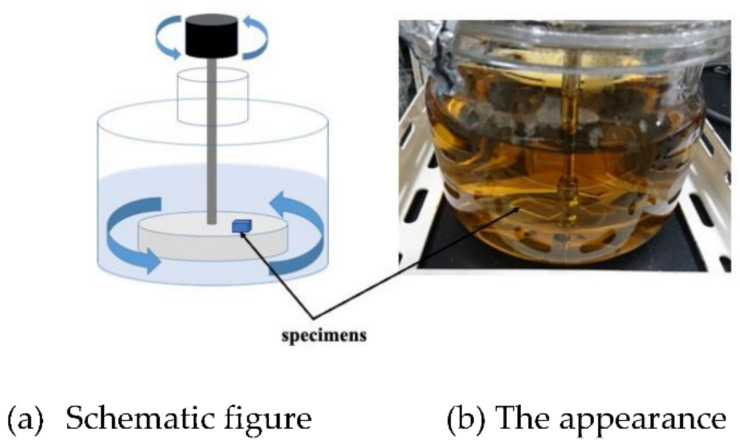
Rotating-platform-driven LBR.

**Figure 3 materials-15-04691-f003:**
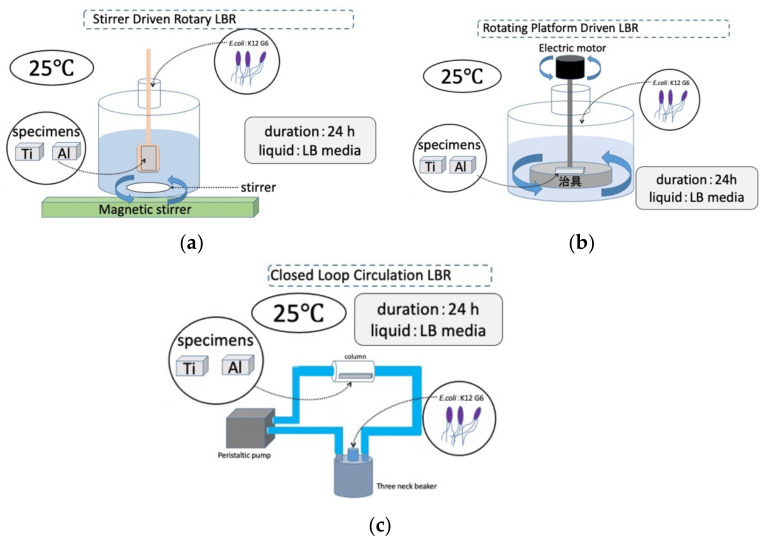
Three kinds of LBRs used in this study and their experimental conditions: (**a**) stirrer-driven rotary biofilm reactor; (**b**) rotating-platform-driven biofilm reactor; (**c**) closed-loop circulation biofilm reactor.

**Figure 4 materials-15-04691-f004:**
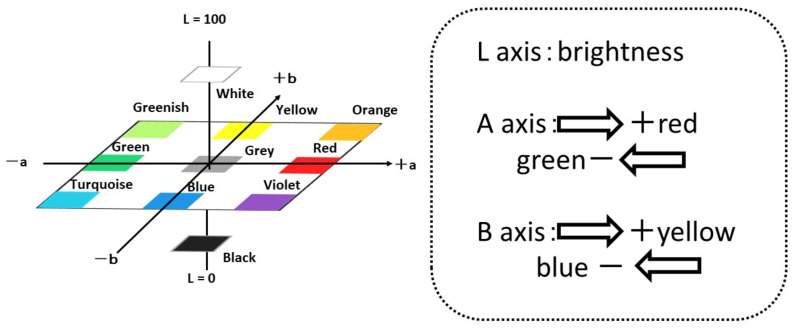
L-a-b color space and mutual relations among various colors.

**Figure 5 materials-15-04691-f005:**
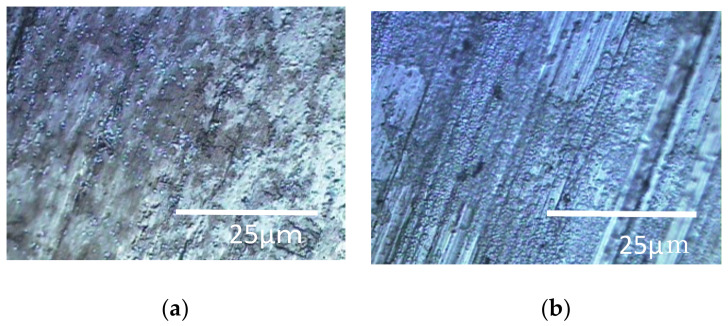
Specimens’ surfaces observed by the optical microscope in the SDRBR: (**a**) titanium specimen and (**b**) aluminum specimen.

**Figure 6 materials-15-04691-f006:**
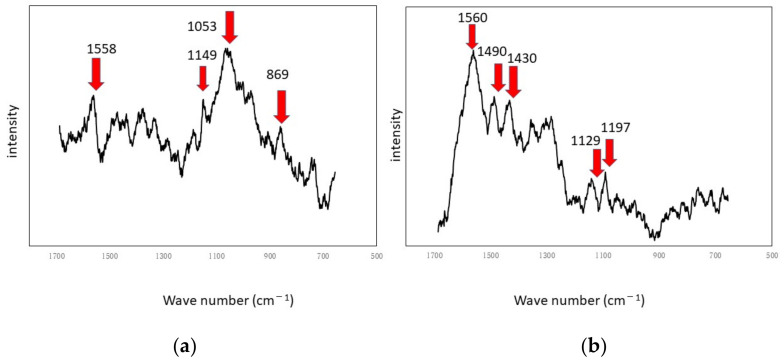
Raman shifts by the SDRBR: (**a**) titanium specimen and (**b**) aluminum specimen.

**Figure 7 materials-15-04691-f007:**
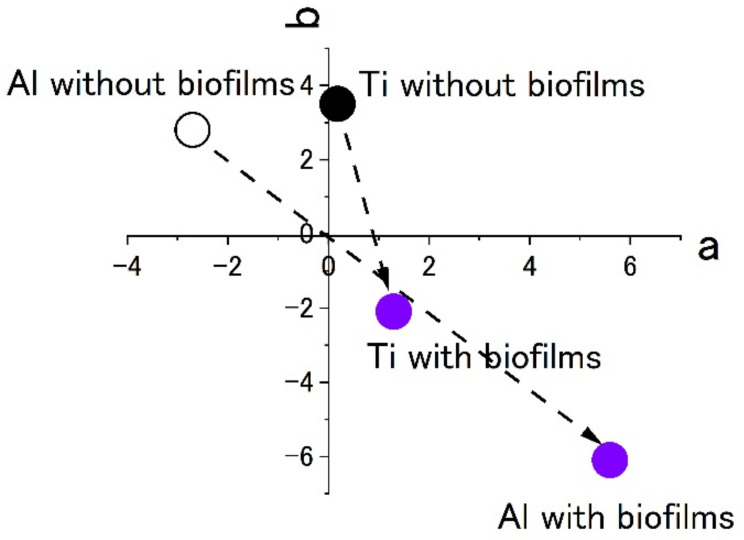
The color changes in specimens’ surfaces stained by 0.1% crystal violet in the SDRBR. (a) complementary color dimension between red and green. (b) complementary color dimension between yellow and blue.

**Figure 8 materials-15-04691-f008:**
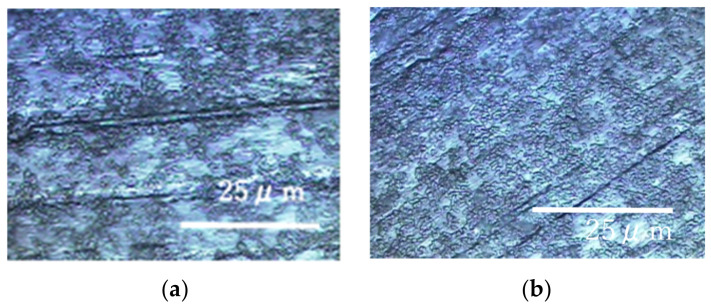
Specimens’ surfaces observed by the optical microscope in the RPDLBR: (**a**) titanium specimen and (**b**) aluminum specimen.

**Figure 9 materials-15-04691-f009:**
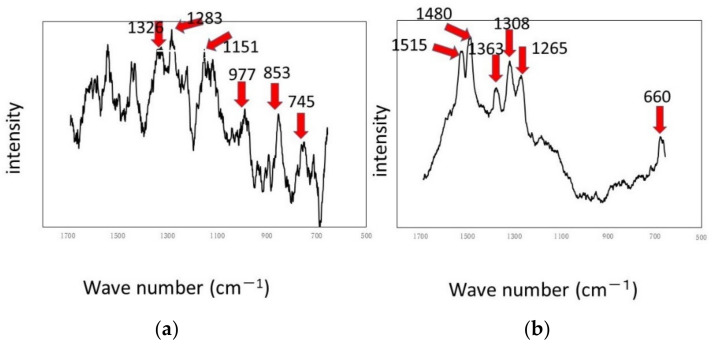
Raman shifts by the RPDLBR: (**a**) titanium specimen and (**b**) aluminum specimen.

**Figure 10 materials-15-04691-f010:**
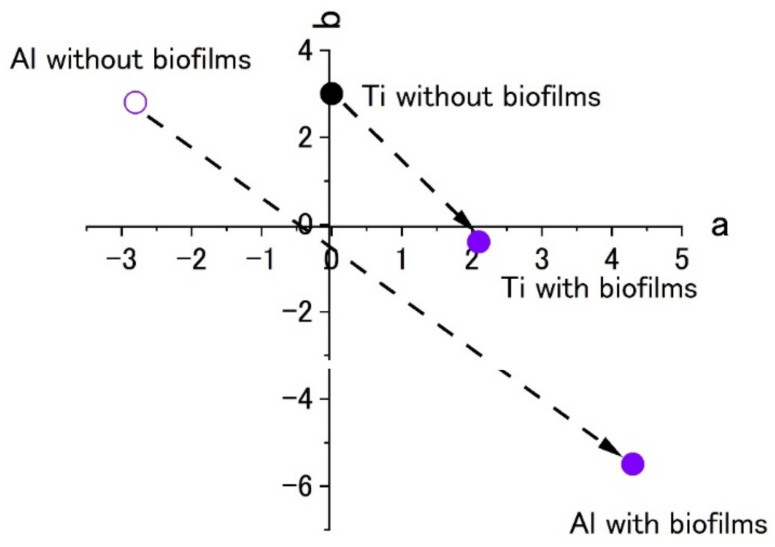
The color changes in specimens’ surfaces stained by 0.1% crystal violet in the RPDLBR. (a) complementary color dimension between red and green. (b) complementary color dimension between yellow and blue.

**Figure 11 materials-15-04691-f011:**
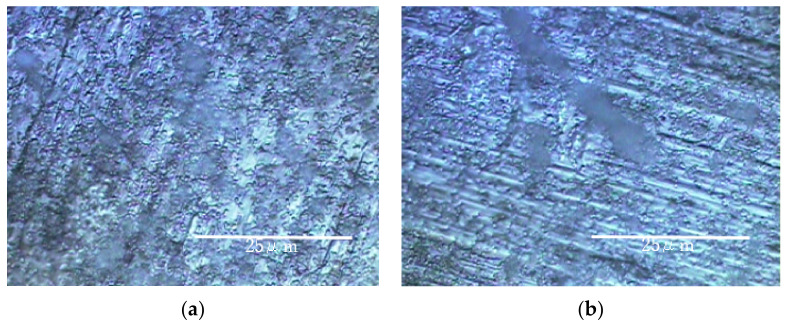
Specimens’ surfaces observed by the optical microscope in the CLCLBR: (**a**) titanium specimen and (**b**) aluminum specimen.

**Figure 12 materials-15-04691-f012:**
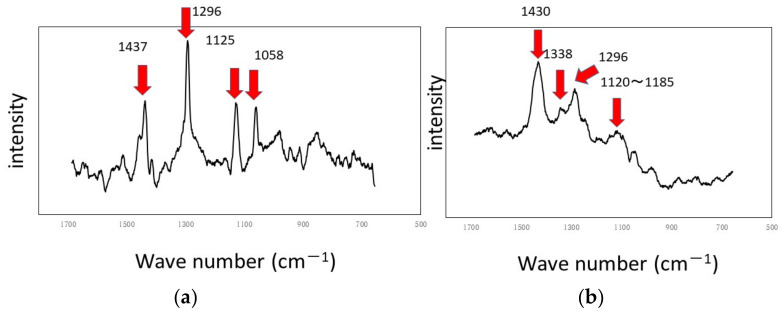
Raman shifts by the CLCLBR: (**a**) titanium specimen and (**b**) aluminum specimen.

**Figure 13 materials-15-04691-f013:**
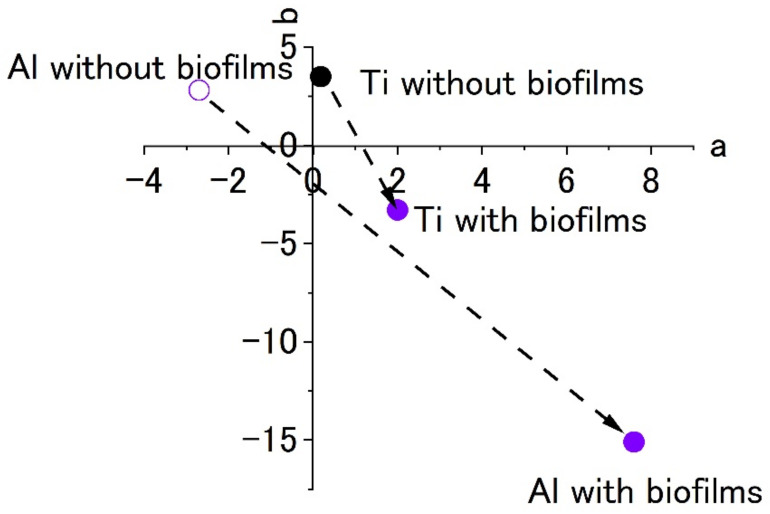
The color changes in specimens’ surfaces stained by 0.1% crystal violet in the CLCLBR. (a) complementary color dimension between red and green. (b) complementary color dimension between yellow and blue.

**Table 1 materials-15-04691-t001:** The comparisons of characteristics among three kinds of LBRs from the practical viewpoint.

LBR	Biofilm Formation	Remarkable Components	Capacity	Ease of Use
SDRBR	medium	proteins	low	simple
RPDLBR	weak	proteins	medium	medium
CLCLBR	strong	Lipids	medium	hard

## Data Availability

Not applicable.

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
