# Peer review of "Proposal for Some Affordable Laboratory Biofilm Reactors and Their Critical Evaluations from Practical Viewpoints"

_materials, 2022, doi:10.3390/ma15134691_

Round 1

Reviewer 1 Report

The present study aims at compare three distinct LBR from their hydrodynamic properties. E. coli was chosen as a model bacterium to evaluate their respective potentials to grow biofilm on metallic substratum that are known to prevent bacterial attachment.

This study, as presented, features many gaps

Two type of specimens are named in the introduction Ti ,Al for preventing biofilm formation, Iron for the opposite. No data are presented for iron !

From introduction and conclusion, it is not clear whether early adhesion or mature biofilm formation is aimed. 24 hours culture is much too short for the second.
Biofilm formation : authors themselves seem to doubt whether biofilm are formed or not. From Raman spectra (with so poor fig. quality), it is strictly impossible to compare and evaluate relative developments of biofilms. More, red arrows and assignments are very fuzzy. The least is to provide exact peak positions with references for their assignment. How one may be sure that “organic compounds” do not come from LB medium ? Is there some negative control (24 h without any inoculum) ?

Violet pictures : photos with specimen before and after culture are needed since the photo quality is very poor.

There are sounding methods to quantify biofilm formation like fluorescence microscopy (dead/live kits). Regarding this point, it is not known whether specimens have been rinsed before freeze drying to remove unattached bacteria.

At which OD600 were inoculated cells ? (and : 500 µm or 500 µL ?)

That the hydrodynamic properties of the three LBR are so badly known restrain the utility of the study.
minor :

Fig 3 and 4 must integrate a “(b)”
some sentence are incomprehensible

The most important is that comparison between LBR is impossible from the data here presented.

Author Response

Dear esteemed first reviewer:

Thank you very much for your valuable time and great comments.  

Let me explain to you about my modification as follows.

(1) E. coli was chosen as a model bacterium to evaluate their respective potentials to grow biofilm on metallic substratum that are known to prevent bacterial attachment.

  No, the reason why we chose E.coli was the abundant experiences and data base. Other researchers used this bacteria and we also used the bacteria in the past. But such a confusion and misunderstanding should be attributed to the lack of explanations and appropriate description in the previous manuscript.  Therefore, we added the sentences from line 112 – 114 newly. 

(2)This study, as presented, features many gaps.

  We checked all of the sentences and also confirmed many gaps.  Therefore, we checked all sentences and tried to eliminate all gaps.  Every correction and modification were mentioned and the comments were given there. 

(3) From introduction and conclusion, it is not clear whether early adhesion or mature biofilm formation is aimed. 24 hours culture is much too short for the second.

    As the reviewer pointed out, the purpose of this study was not clear in the previous manuscript.  Therefore, it was very natural for the reviewer to ask such a question.  Therefore, we revised the introduction part and tried to make clear that the purpose of this study was the comparison among three laboratory biofilm reactors from the practical viewpoint, because we developed three kinds of LBRs for the social implementation.  This journal’s original aim was also to collect the social implementation topics relating to materials’ science.  One of the practical viewpoints is rapid evaluation.  Therefore, we wanted to check the biofilm at the initial stage and the comparison among them then.  24 hours culture was short, but enough for our purpose this time.

(4) Two type of specimens are named in the introduction Ti ,Al for preventing biofilm formation, Iron for the opposite. No data are presented for iron !

Yes, as the reviewer pointed out, iron’s data should have been deleted.  Iron tends to form biofilms too much.  And according to our study, the biofilm usually turns out to be scales or corrosion products.   According to our research in the past, the biofilm was changed to corrosion products in about 6 hours under a certain condition.  The process is usually continuous.  Because we wanted to check the biofilm formation at the initial stage. And we didn’t intend the structural comparison of biofilms on various specimens. And we wanted to avoid the involvement of corrosion products found on iron specimens.  Therefore, we deleted the iron results this time. 

(5) Biofilm formation : authors themselves seem to doubt whether biofilm are formed or not.

  Such a description giving readers was inappropriate.  Therefore, we modified the explanations for Raman peaks. We adopted finger print method and compared our results with the data in the past.  Then we fixed and presumed the EPS by using Raman peaks. 

(6) From Raman spectra (with so poor fig. quality), it is strictly impossible to compare and evaluate relative developments of biofilms.

Yes, you are right.  However, that was not our purpose.  Moderate identification was enough for this study.  Plus, we checked the specimens by staining.  Usually, biofilms identification should be checked not only by one method, but also by plural methods.  To make our purpose and intention clearer, we revised sentences and also tried to identify peaks by finger print methods.  Finger print methods could not identify the peaks strictly, but such a strict identification was not originally our goal.  I guess our not so good description in the previous manuscript must mislead you, so I believe.  Therefore, we revised and corrected some places shown in the manuscript. 

(7) More, red arrows and assignments are very fuzzy. The least is to provide exact peak positions with references for their assignment.

As shown in the manuscript, we added the peaks values concretely and showed the identification results and corresponded references based on the data in the past (Finger print method). 

(8) How one may be sure that “organic compounds” do not come from LB medium ? Is there some negative control (24 h without any inoculum) ?

This question is essential in a meaning.  However, it is hard to answer correctly.  Organic compounds might come from the LB medium partly.  However, even if such a compound constituting the medium, the interpretation was not changed.  Such compound was also considered one of biofilm components, so long as it would be attached to materials’ surfaces firmly.  Due to the existence of biofilms on materials’ surfaces, the surface becomes sticky and any organic and inorganic compounds sticked to materials’ surfaces firmly and detected, so we could presume.  Therefore, the derivation of organic compounds are not so important problems to identify biofilms entirely.  Besides, we identified many biofilm peaks using the references in the past.  Particularly the reference 38 (Cao’s paper) could mention many peaks from some representative bacteria (Pseudomonas putida, E.coli etc.) and their biofilms.  Therefore, we believe our results have shown the existence of biofilms directly and indirectly. 

(9) Violet pictures : photos with specimen before and after culture are needed since the photo quality is very poor.

We are so sorry that those pictures were not so sharp.  However, the photos showing real colors could not be taken so easily.  On the other hand, we felt that these photos showed our limitations to take pictures clearly.  Therefore, we changed the pictures from photos to the color measurement values. (Figures 7, 10, 13).  To explain about the color space, we added Figure 4 in the experimental procedure part. 

(10) There are sounding methods to quantify biofilm formation like fluorescence microscopy (dead/live kits). Regarding this point, it is not known whether specimens have been rinsed before freeze drying to remove unattached bacteria.

Fluorescence microscopy is also one of them.  Actually, there are many methods using relatively expensive methods.  For fundamental studies, those methods might be helpful.  However, the main point of this current study is not to identify biofilms by some bacteria, nor to investigate the bacterial structures.  In addition to that, biofilms were not compatible of the number of bacteria.  And we have investigated biofilms using Raman spectroscopy so far, as mentioned references from references 26 to 34.  We could compare the current results with the ones in the past.  Therefore, we didn’t use the method this time.  However, your points have a meaning in other study.

As for rinsing, we didn’t rinse specimens specially.  However, the specimens serving to Raman spectroscopy were prepared through many repeated immersion steps into alcohol solutions.  Therefore, compounds or bacteria just attaching to materials surfaces without any special bonding were removed easily.  The process was described from line 148 to 156. 

(11) At which OD600 were inoculated cells ? (and : 500 µm or 500 µL ?)

We don’t understand the meaning of this question.  The OD of inoculated cells didn’t have any relation to the plot of this experiment. 

(12) That the hydrodynamic properties of the three LBR are so badly known restrain the utility of the study.

The analysis of hydrodynamic properties were actually very difficult and such an analysis doesn’t lead to our goal directly.  Crystal staining method is now going to be an international standard to evaluate biofilms on materials.  The standardization activity is the collaboration between SIAA (mentioned in our acknowledgment) and me actually and from the same practical viewpoint, we were interested in the invention of LBR from the practical viewpoint. The evaluation should have been done from the viewpoint also.   We originally wanted to write this paper for the purpose and also the social implementation.  Therefore, we added the new table 1 to make it clear. 

(13) Fig 3 and 4 must integrate a “(b)”

Because we revised the original figures, the problem was solved, so we believe. 

(14) The most important is that comparison between LBR is impossible from the data here presented.

     The description in the previous manuscript was not appropriate and therefore, it gave the esteemed reviewer such an impression.  To improve this, we revised many parts which we showed in the text directly.  Hopefully, this revision will give him/her the different impression. 

Reviewer 2 Report

The approached subject has practical applicability towards developing and / or recommending of antibiofilm materials, with benefits for various industries which should be pointed out.

Please avoid abbreviations in the title.

Abstract should be improved with some clear information on the context of the study, the aim, the main results and how to apply them in practice.

The conceptualization of this study is quite modest, no solid results are provided in support of the claims.

Introduction needs to be significantly improved. There is a lack of essential information on the state of knowledge and some results obtained by scientists in this regard. The contribution of the authors, the novelty and the purpose of this study should be emphasized. Bibliographic references are not properly cited; they should be analyzed individually but not grouped as [1-7]; [8-33].

The methodology and materials section also needs to be improved. Section 2.1 lacks important details regarding the materials, their main characteristics or properties, the supplier, etc. There are many deficiencies in expression, the paper must be completely revised and written in a professional style. There are also lots of typos (e.g. use Italics for E. coli; subscript in H2O; there are wide spaces between many sentences). Some sentences need to be completely reworded, e.g. <The third type is called “Closed Loop Circulation LBR” (CLC LBR) among us authors tentatively>. In section 2.4 the first person was used, it needs to be corrected. In section 2.4.2 the authors used the present tense instead of the past tense.

Discussions are weak, they need to be more consistent. Abbreviations should be used only when first appearing in the text, not in the Discussions too.

Conclusions are not structured based on the results; they are rather a summary of the experimental study. The study of documentary material carried out by the authors must be reflected in the text, mainly in the Introduction and Discussions.

Author Response

Dear the esteemed second reviewer:

Thank you so much for your kind and great tips and comments.  We appreciate you very highly.  

(20) Please avoid abbreviations in the title.

We avoided abbreviations in the title and also modified the title, so that it would be compatible with the contents.

(21) Abstract should be improved with some clear information on the context of the study, the aim, the main results and how to apply them in practice.

We revised the abstract entirely, according to the reviewer’s tips, as much as possible. 

(22) Introduction needs to be significantly improved. There is a lack of essential information on the state of knowledge and some results obtained by scientists in this regard. The contribution of the authors, the novelty and the purpose of this study should be emphasized.

We revised the introduction according to the reviewer’s comments as much as possible. 

(23) Bibliographic references are not properly cited; they should be analyzed individually but not grouped as [1-7]; [8-33].

I am sorry that the reviewer had such an impressions.  We don’t think always that the grouped references were so bad, because there were many kinds of viewpoints for a refence topic.  In this case, the references from 1 to 7 belong to this category.  However, we would like to make much of reviewer’s idea.  Therefore, we divided the references as much as possible, but please allow us not to give each reference per each topic. 

(24) The methodology and materials section also needs to be improved. Section 2.1 lacks important details regarding the materials, their main characteristics or properties, the supplier, etc.

Experimental part was revised, according to the reviewer’s tips. 

(25) There are many deficiencies in expression, the paper must be completely revised and written in a professional style. There are also lots of typos (e.g. use Italics for E. coli; subscript in H2O; there are wide spaces between many sentences). Some sentences need to be completely reworded, e.g. <The third type is called “Closed Loop Circulation LBR” (CLC LBR) among us authors tentatively>.

We believe all were revised according to the reviewer’s comments.  Thank you!

(26) In section 2.4 the first person was used, it needs to be corrected. In section 2.4.2 the authors used the present tense instead of the past tense.

The original sentences were modified pretty much.  Therefore, the entire sentences were revised totally.

(27) Discussions are weak, they need to be more consistent.

    Discussion part was revised significantly. 

(28) Abbreviations should be used only when first appearing in the text, not in the Discussions too.

  I don’t understand this, unfortunately.  Abbreviations should be used positively after the words appear the second time or later. 

(29) Conclusions are not structured based on the results; they are rather a summary of the experimental study. The study of documentary material carried out by the authors must be reflected in the text, mainly in the Introduction and Discussions.

We revised the conclusions according to the tips.  Thank you for your great tips!

Round 2

Reviewer 1 Report

the amendments fulfill reviewers requirements

Reviewer 2 Report

The manuscript has been substantially improved according to my suggestions. I agree with the authorts' comments on the citation from multiple sources.